# Typical and Aberrant Functional Brain Flexibility: Lifespan Development and Aberrant Organization in Traumatic Brain Injury and Dyslexia

**DOI:** 10.3390/brainsci9120380

**Published:** 2019-12-16

**Authors:** Stavros I. Dimitriadis, Panagiotis G. Simos, Jack Μ. Fletcher, Andrew C. Papanicolaou

**Affiliations:** 1Division of Psychological Medicine and Clinical Neurosciences, School of Medicine, Cardiff University, Cardiff CF14 4XN, UK; 2Cardiff University Brain Research Imaging Centre, School of Psychology, Cardiff University, Cardiff CF24 4HQ, UK; 3School of Psychology, Cardiff University, Cardiff CF10 3AT, UK; 4Neuroinformatics Group, Cardiff University Brain Research Imaging Centre, School of Psychology, Cardiff University, Cardiff CF24 4HQ, UK; 5Neuroscience and Mental Health Research Institute, Cardiff University, Cardiff CF24 4HQ, UK; 6MRC Centre for Neuropsychiatric Genetics and Genomics, School of Medicine, Cardiff University, Cardiff CF24 4HQ, UK; 7School of Medicine, University of Crete, Herakleion 70013, Greece; akis.simos@gmail.com; 8Institute of Computer Science, Foundation for Research and Technology, Herakleion 70013, Greece; 9Department of Psychology, University of Houston, Houston, Texas, TX 77204-5022, USA; Jack.Fletcher@times.uh.edu; 10Division of Clinical Neurosciences, Department of Pediatrics, University of Tennessee Health Science Center, Memphis, TN 38103, USA; apapanic@uthsc.edu; 11Le Bonheur Neuroscience Institute, Le Bonheur Children’s Hospital, Memphis, TN 38103, USA

**Keywords:** resting-state activity, magnetoencephalography, brain maturation, cross-frequency coupling, reading disability, traumatic brain injury

## Abstract

Intrinsic functional connectivity networks derived from different neuroimaging methods and connectivity estimators have revealed robust developmental trends linked to behavioural and cognitive maturation. The present study employed a dynamic functional connectivity approach to determine dominant intrinsic coupling modes in resting-state neuromagnetic data from 178 healthy participants aged 8–60 years. Results revealed significant developmental trends in three types of dominant intra- and inter-hemispheric neuronal population interactions (amplitude envelope, phase coupling, and phase-amplitude synchronization) involving frontal, temporal, and parieto-occipital regions. Multi-class support vector machines achieved 89% correct classification of participants according to their chronological age using dynamic functional connectivity indices. Moreover, systematic temporal variability in functional connectivity profiles, which was used to empirically derive a composite flexibility index, displayed an inverse U-shaped curve among healthy participants. Lower flexibility values were found among age-matched children with reading disability and adults who had suffered mild traumatic brain injury. The importance of these results for normal and abnormal brain development are discussed in light of the recently proposed role of cross-frequency interactions in the fine-grained coordination of neuronal population activity.

## 1. Introduction

The study of human brain development is rapidly becoming a central research area for understanding the nature of neuropsychiatric diseases and various developmental disorders [1,2]. A major breakthrough in this line of research was the demonstration of coherent patterns of brain activity at rest [3,4,5], advancing the notion that the human brain is a self-organizing system constantly displaying coherent patterns of activity, both locally and globally, rather than a passive device solely driven by bottom-up processes [6,7,8]. A key development in this line of research was the demonstration of intrinsic connectivity networks such as the salience, prefrontal, sensorimotor, and default mode networks, derived from functional connectivity analyses of resting-state functional magnetic resonance imaging (fMRI) data.

Several developmental trends have since been described including changes in the strength of short- and long-range connections [9], the expansion of cortical hubs outside sensorimotor regions [10], and increasing variability of connections between the default mode, visual, and cerebellar networks [11]. Significant improvements in algorithms used to quantify functional connectivity have created realistic expectations towards a better understanding of neurophysiological correlates of brain maturation as well as identifying robust brain connectivity markers of individual developmental trajectories [9,12]. Previous resting-state fMRI studies based on static brain connectivity and multivariate pattern analysis tools (MVPA, support vector machine, and support vector regressor) attempted to predict individual age [9,13] and classify individual participants according to their actual age [13,14]. A dynamic functional connectivity study based on fMRI resting-state further demonstrated that the temporal variability in the strength of specific connections afforded more accurate modelling of spontaneous fluctuations related to maturation age [11]. 

The vast majority of developmental neuroimaging studies have used fMRI to model functional connectivity patterns relying on indirect associations between rhythmic fluctuations in resting-state hemodynamic recordings and underlying neurophysiological activity. Few studies have employed brain electrical or neuromagnetic recordings operating at a time scale suitable to accurately represent the rhythmic patterns of neurophysiological activity at both low- (e.g., in the δ band ranging between 1 and 4 Hz) and higher frequencies (θ: 5–8 Hz, α: 9–12 Hz, β: 13–30 Hz, and γ: 30–100 Hz; [6,8]). 

Recently, two basic dominant intrinsic coupling modes (dICMs) have been documented, one indexed by the correlation of the amplitude envelope and the second by phase synchronization [15]. Each dICM purportedly displays a characteristic, complex spectral and spatial signature undergoing systematic long-term (i.e., developmental) changes [15]. A recent study on 59 participants aged 6–34 years reported age-related increases in the magnitude of inter-regional correlations in α and β frequency bands supporting the notion of developmental growth of the degree of neurophysiological integration both locally and globally [16]. Moreover, simulated neuromagnetic recordings highlighted the role of delayed network interactions involving amplitude envelope coupling in various frequency bands in the emergence of spontaneous functional connectivity [17]. An important development in the study of dICMs has been the demonstration of cross-frequency coupling as a mechanism supporting communication of neuronal populations operating at different dominant frequencies at rest [18]. Evidence of the clinical significance of cross-frequency coupling has been presented by our group in patients who had suffered mild traumatic brain injury [19] and in children with dyslexia [20]. Furthermore, there is growing evidence that the degree of short-term variability in functional connectivity profiles at rest (i.e., during the course of the recording session) may serve as a phenotypic characteristic of certain mental disorders such as autism, Attention Deficit Hyperactivity Disorder, and schizophrenia [21]. During normal development, higher levels of non-stationarity in functional connectivity, at least within a certain range, may underlie the capacity of brain networks to adapt to changing external demands [22]. 

In a recent study, we demonstrated the complexity of activity and brain connectivity in functional neuroimaging under the notion of dICM [23,24,25,26,27,28,29]. We defined a novel flexibility index (FI) tailored to EEG, MEG, and fMRI timeseries that quantifies the rate of transition from one dICM to another in consecutive temporal segments for every pair of timeseries. This index was highly reproducible over repeated scan sessions [29].

The present work utilizes a variety of dynamic functional connectivity indices computed on resting-state, sensor-level neuromagnetic data from a large cross-sectional cohort (*N* = 178; by collapsing data across two MEG systems) of healthy volunteers aged 8–60 years to model age-related individual differences. The multi-step analytic method adopted here sought to identify dominant (i.e., statistically significant and topologically salient) types of coupling between underlying neuronal populations as they evolve in time during the recording session. We examined a comprehensive set of measures of intra- and cross-frequency coupling of potential neurophysiological relevance to address the following primary goals: (1) identify characteristic dICMs within and between lobar regions that demonstrate systematic developmental trends; and (2) develop a measure of temporal variability in dICMs, integrated across the entire network of MEG sensors, which could serve as a reliable indicator of participant age. Secondary goals of the present work were: (i) to assess the sensitivity of the aforementioned indicator in differentiating between typical and atypical brain function (in groups of children with dyslexia and adults who had suffered mild traumatic brain injury); (ii) to assess the reproducibility of this indicator across repeated scan sessions; and (iii) to evaluate the equivalence of results related to age-prediction across MEG systems.

## 2. Material and Methods

### 2.1. Participants

The principal dataset consisted of resting-state MEG data (eyes-closed) from 178 right-handed participants without history of neuropsychiatric disease, sensory deficit, or learning disability, who were assigned to six age groups: 8–12 (*n* = 24, 24 men), 13–17 (*n* = 26, 26 men), 18–27 (*n* = 43, 13 men), 28–37 (*n* = 43, 11 men), 38–50 (*n* = 28, 20 men), and 51–60 (*n* = 14, three men) years. Data from 81 healthy participants were obtained with a 248-channel Magnes WH3600 system (4D Neuroimaging Inc., San Diego, CA; Magnes-248) equipped with 248 first-order axial gradiometer coils at the University of Texas Health Science Center. Data from the remaining 97 healthy participants were drawn from the OMEGA, Open MEG Archive and were obtained in two identical 275-channel CTF systems (VSM MedTech Inc., Coquitlam, BC, Canada; CTF-275) located at the McConnell Brain Imaging Centre of the Montreal Neurological Institute, McGill University and at the Université de Montréal [30].

Additional resting-state datasets (eyes-open) were obtained from (a) 10 healthy right-handed young adults (five women, aged 24.4 ± 1.5 years) on two occasions to assess test–retest reliability of dynamic functional connectivity indices, (b) 25 right-handed children with reading disability (13 girls, aged 12.2 ± 3.1 years), as indicated by scores below the 16th percentile level (standard score of 85) on the Basic Reading composite index (average of Word Attack and Letter–Word Identification subtest scores of the Woodcock–Johnson Tests of Achievement-III; for additional details on recruitment and participant characteristics see [31]), and (c) 30 right-handed adults who had suffered mild traumatic brain injury (mTBI; 12 women, averaging 29.3 ± 9.2 years of age. Inclusion criteria for mTBI patients required the presence of a head injury occurring within the preceding 24 h, Glasgow Coma Scale score 13–15, loss of consciousness 0–30 min, post-traumatic amnesia 0–24 h, and a negative head CT scan; for additional details on recruitment and participant characteristics see [19]). These three datasets were acquired on a Magnes-248 system at the University of Texas Health Science Center.

Participants in the reliability study were master’s students at the School of Psychology, Cardiff University. Test–retest resting-state recordings (eyes-open) were obtained on a CTF-275 system at Cardiff University. 

All participants signed the related consent form and this pilot study was approved by the ethical committee within the school.

### 2.2. Data Recording and Preprocessing

Magnes-248 data were collected at a sampling rate of 1017.25 Hz and online bandpass filtered between 0.1–200 Hz for 3 min. CTF-275 data were collected at a sampling rate of 2.400 Hz for 10 min from which the first 3 min were used here. Recordings were downsampled to 170 Hz (Magnes-248) or 150 Hz (CTF-275), resulting in 30,600 and 27,000 sampling points, respectively. Analyses were performed in the native sensor space for each system.

Preprocessing entailed artefact reduction using independent component analysis, conversion to planar gradiometer field approximations, and bandpass-filtering in the following frequency ranges using a 3rd-order Butterworth filter (in zero-phase mode): 0.5–4, 4–8, 8–10, 10–13, 13–15, 15–19, 20–29, and 30–45 Hz corresponding to δ, θ, α_1_, α_2,_ β_1_, β_2_, β_3_, and γ bands. Additional preprocessing details are provided in Appendix A, Section 1.

### 2.3. Dynamic Functional Connectivity

The goal of the first step of the analytic procedure was to capture dynamic functional connectivity in the form of distinct within- and cross-frequency coupling modes based on amplitude and phase. This was achieved by computing five complementary indices of signal coupling: amplitude envelope correlation (AEC), phase-to-amplitude cross-frequency coupling (CFC^PAC^), intra-frequency phase-to-phase coupling, intra and cross-frequency delay symbolic transfer entropy (dSTE), and directed phase lag (dPLI). Furthermore, data-driven statistical thresholding was employed to identify pair-wise (sensor-to-sensor) connectivity values that were unlikely to have occurred by chance alone. 

#### 2.3.1. Intra and Inter-Frequency Coupling Estimators and Statistical Filtering

Each set of connectivity indices (for each frequency and/or between two frequencies recorded at the same or across different sensors, when appropriate) was computed independently within 2-s time windows using a sliding window approach. The width of the temporal window was set equal to the duration of two cycles of δ activity (i.e., 2 s) ensuring that modulations of activity by the lowest frequency band (δ) would be preserved when estimating cross-frequency coupling. Unless otherwise specified below, the statistical significance of each connectivity index value was assessed using corresponding values derived from 10,000 surrogate time-series.

Connectivity indices were computed for each surrogate dataset and the probability that a given observed connectivity index value could belong to the corresponding surrogate distribution was estimated. This probability reflected the proportion of surrogate connectivity values that were higher than the observed index value [5]. The False Discovery Rate (FDR) method [32] was employed to control for multiple comparisons (across intra-frequency coupling and all possible pairs of frequencies) with the expected proportion of false positives set to *p* = 0.01. For further computational details, see the Appendix A, Section 2.

#### 2.3.2. Amplitude Envelope Correlation (AEC)

The time courses of bandpass-filtered magnetic activity at each sensor were Hilbert transformed and the resulting absolute amplitude value was used to compute the envelope of oscillatory power in each frequency band. The Hilbert envelope technique has been used extensively in previous MEG studies (for a mathematical description see [33]). At each temporal segment, the amplitude envelope correlation represented Pearson correlation coefficients computed on the Hilbert envelopes to assess coupling between sensors, either in the same or different frequency bands. Surrogate data analyses were conducted to retain non-chance correlations. The final AEC (binary) dataset identified the frequency(-ies) of the highest, significant correlations for a given pair of sensors.

#### 2.3.3. Phase-to-Amplitude Cross-Frequency Coupling (Cross-Frequency iPLV)

Cross-Frequency Coupling (CFC) analyses were implemented to identify the prominent pair of interacting frequencies, both between and within sensors [34,35,36]. Among the available CFC descriptors, phase-amplitude coupling (PAC), which relies on phase coherence, is the one most commonly encountered in research [37,38], adapted to continuous MEG multichannel recordings [23,39,40,41]. 

The PAC mode that characterized a specific pair of frequencies was determined based on the highest, statistically significant PAC value in the surrogate analyses. The dominant PAC values for each pair of sensors and across sliding windows were integrated across frequency bins yielding 28 possible pairwise PAC estimates among the eight frequency bands. 

#### 2.3.4. Intra-Frequency Phase-to-Phase Coupling (Same-Frequency iPLV)

Computation of intra-frequency phase-to-phase coupling represents a special case of the procedure described above to compute iPLV where the two signals are of the same frequency. Intra-frequency phase coupling was estimated using the Hilbert phase transform. Based on surrogate data analyses, we identified the dominant iPLV values for each pair of sensors and across sliding windows. 

#### 2.3.5. Cross-Frequency Interactions via Delay Symbolic Transfer Entropy (dSTE)

Symbolic transfer entropy was proposed to overcome the limitations of optimized parameters required for estimation of transfer entropy [42]. In the present study, we adopted the neural gas algorithm (NG; [43]) to create a common codebook for the entire set of sensor pairs [23,24,44,45]. Furthermore, significant causal interactions between two sensors, A and B, were identified by applying an adaptation of transfer entropy for symbolic time series [46,47,48,49]. 

The final dSTE dataset contained the strength, direction, and delay of the significant and dominant pair of frequencies for each sensor [23,24,45] as derived from the surrogate data analyses. If more than two dSTE frequencies or frequency pairs were significant, the one with the maximum dSTE value was selected. 

#### 2.3.6. Phase interactions: Directed Phase Lag Index (dPLI)

The directed phase lag index [50] was employed to assess potential causal relationships based on the phase difference between two oscillations either in the same or different frequencies. The highest, significant dPLI values for each pair of sensors, frequencies, and sliding windows was identified via surrogate analyses. 

#### 2.3.7. Identifying the Dominant Intrinsic Coupling Mode (dICM) for A Given Pair of Sensors

The steps described in Section 2.3.2, Section 2.3.3, Section 2.3.4, Section 2.3.5 and Section 2.3.6 resulted in three arrays for each of the five connectivity estimators. The first array contained the AEC, cross-frequency iPLV, dSTE, same-frequency iPLV, or dPLI values, and the second array contained the corresponding *p*-values. In the third array of size (2 × 900 × 248/275 × 248/275), the identity of the dominant coupling mode (based on AEC, cross-frequency iPLV, dSTE, same-frequency iPLV, or dPLI values) was indicated by a numeric (integer) code. 

Next, we selected a single representative connectivity estimator for each sensor pair and time window (dICM). When considering multiple estimators, the statistical threshold was reset to *p* < 0.01/4 = 0.0025. If more than one connectivity estimator exceeded this threshold, they were both maintained as representative dICM for this particular pair of MEG sensors and temporal segment. This information was stored in an array of size (2 × 900 × 248/275 × 248/275) where the 1st dimension codes the identity of the connectivity estimator (1–5) and the second dimension reflected the frequency of the signals (e.g., α, α → β) coded by integer values (1–36). As described in detail in Section 2.3.9, dICMs were subsequently integrated across groups of sensor pairs in each hemisphere. The process of identifying the dICM for every pair of MEG sensors is schematically illustrated in Figure 1.

#### 2.3.8. Topological Filtering based on Orthogonal Minimal Spanning Tress (OMSTs)

After applying statistical filtering to identify characteristic dICMs per pair of MEG sensors, topological filtering was employed in order to define meaningful network structure [51]. A data-driven topological filtering scheme was adopted utilizing orthogonal minimal spanning trees (OMSTs [25,26]). This iterative method relies on the weights of the connections within a network and their topology to optimize information flow (as defined by global efficiency) and minimize the cost of the surviving connections. 

#### 2.3.9. Identifying the Dominant Type of Inter- and Intra-Hemispheric Interactions for Groups of Neighbouring Sensors

Significant connectivity values for each sensor pair (contained in the 2-dimensional array described in Section 2.3.7) were subsequently integrated over groups of neighbouring sensors roughly corresponding to underlying lobar anatomy. In this manner, we determined the dominant type of interaction (dICM) between hemispheres for a given “lobe” (cross-hemispheric) and between lobes for both hemispheres (within-hemisphere) as described below. 

The characteristic dICM between “lobes” for a given participant was identified as the interaction mode demonstrated by >75% of the sensors between two lobar sectors and by at least 50% of all temporal segments. At the group level, the dominant type of interaction between hemispheres for a given lobe, or between lobes for a given hemisphere, was determined as the dominant connectivity mode displayed by all participants in a particular age group. Over the entire sensor array and for each participant, the predominance of a given interaction mode was quantified using two complementary indices: the mean subgraph strength (MSS) and fractional occupancy (FO). 

MSS reflected the average strength of interactions that were found to characterize signal interdependencies either between the sensors comprising a given lobar sector of the sensor array (“lobe”), or between two such sectors. The MSS index describing sensor interactions between two “lobes” was defined as follows:(1)$${\mathrm{MSS}}_{\mathrm{LOBES}}=\frac{\sum_{t=1}^{\mathrm{temporal}\;\mathrm{segments}\;}\frac{\sum_{n1=1}^{\mathrm{sensors}1}\sum_{n2-1}^{\mathrm{sensors}2}\mathrm{FCG}\left(s1,s2\right)\;\;}{\mathrm{sensors}1⥂\;x\;\mathrm{sensors}2}}{\mathrm{temporal}\;\mathrm{segments}}\;$$
where sensors_1,2_ refers to the total number of sensors defining each “lobe”, and temporal segments refers to the number of time windows where a particular type of dICM was deemed to be the dominant type of interaction. 

The FO index is defined as the ratio of the number of temporal segments where a particular type of dICM was deemed to be the dominant type of interaction between two “lobes” (N^Dom^) divided by the total number of temporal segments (N^TS^):(2)$$\mathrm{FO}=\frac{N^{\mathrm{Dom}}}{N^{\mathrm{TS}}}$$

The MSS and FO values range between 0 and 1. This procedure resulted in 12 MSS indices computed for within- and between-sensor associations, and 10 FO indices, which by definition, were computed only for between-sensor associations. Equations (1) and (2) were adapted to provide indices of within-hemisphere interactions between sensors in a given lobar region. 

#### 2.3.10. Flexibility Index (FI) Based on Dominant Intrinsic Coupling Modes

Finally, a flexibility index was developed in order to quantify temporal variability in dICMs at the level of sensor pairs, integrated over the entire sensor network, as originally proposed by Bassett et al. [52] for EEG and MEG data. FI was computed from the individual 3D matrix of size 900 (temporal segments) × 248/275 (sensors) × 248/275 (sensors) containing the identity label of dominant interaction modes that survived the statistical filtering described in Section 2.3.7. FI reflects the rate of dICM changes between every two consecutive temporal segments for each pair of MEG sensors [28,29]. Integrated over the entire sensor network, FI values range between 0 and 1, according to the formula:(3)$$\begin{array}{l} {\mathrm{FI}}^{\mathrm{MEG}}\left(\mathrm{Sensors},\mathrm{Sensors}\right)\\ =\frac{1}{T-1}\overset{T-1}{\underset{s=1}{\sum }}\overset{\mathrm{Sensors}}{\underset{{\mathrm{sensor}}_{1}=1}{\sum }}\overset{\mathrm{Sensors}}{\underset{{\mathrm{sensor}}_{2}=1}{\sum }}\delta (\mathrm{DICM}\left(T,{\mathrm{sensor}}_{1},{\mathrm{sensor}}_{2}\right),\mathrm{DICM}(T\\ +1,{\mathrm{sensor}}_{1},{\mathrm{sensor}}_{2})) \end{array}\;{\mathrm{FI}}_{\mathrm{GLOBAL}}^{\mathrm{MEG}}=\frac{{\mathrm{FI}}^{\mathrm{MEG}}}{\mathrm{Sensors}\times \mathrm{Sensors}}\;$$
where T = 900.

### 2.4. Modelling Participant Age through Individual FI values

The type of association between individual FI values and participant age (in years) was assessed by fitting a series of models (linear, quadratic, log, exponential, and Von Bertalanffy; [9]. Model comparison was based on the Akaike information criterion (AIC) with smaller values indicating better fit. 

For comparison purposes, two well-established measures of brain activity and sensor interdependence were also computed, namely relative power spectrum and imaginary part of coherence. We also estimated multi-scale entropy that has demonstrated its ability to detect age-dependent differences of brain activity [53]. In contrast to the dICM indices, they provide static representations of the strength of oscillatory activity and within-frequency phase coupling, respectively (for computational details see Appendix A, Section 4 and Appendix A).

### 2.5. Deriving Age-Related Neuromagnetic Features

The differential sensitivity of dICM- and supplementary measure-related features (power spectrum, coherence, multi-scale entropy) to participant age, either as a continuous or as an ordinal variable (age-group), was assessed via support vector regressors and multi-class support vector machines, respectively. Each method was applied to the same set of 12 dICMs represented by 12 MSS and 10 FO values, which in turn, characterized distinct, empirically derived sensor subnetworks, each displaying a distinct topography and dynamic functional connectivity mode (see Table 1). 

Prediction accuracy was cross-validated using leave-one-out and 5-fold procedures on data aggregated across the two MEG-systems, supplemented by across MEG-system cross-validation schemes (i.e., employing data from one MEG system as the training set and data from the second MEG system as the testing set, and vice versa; see Appendix A, Section 5). Analyses on the supplementary measures were conducted for each measure separately (including all salient features that were empirically-derived for each measure) as well as on the entire set of power spectrum, imaginary part of coherence, and multi-scale entropy features combined.

### 2.6. Software for Analyses

All analyses were conducted using custom in-house software in MATLAB (version R2018b, Natick, Massachusetts, USA: The MathWorks Inc.) and Fieldtrip (Donders Institute for Brain, Cognition and Behaviour, Radboud University, The Netherlands) basic routines for reading MEG files.

## 3. Results

### 3.1. Age-Related Differences in dominant Intrinsic Coupling Modes (dICMs)

Table 1 illustrates the 12 dICMs between and within “lobes” and the corresponding MSS and FO values that demonstrated age-dependencies. Concerning the first goal of the study, there were clear developmental trends in the degree of predominance of each of the 12 dICMs, as shown in the lower panels of Figure 2 and Appendix A, demonstrating significant increases in MSS and FO through 28–37 years, followed by a gradual decline thereafter. For instance, Figure 2 demonstrates the dominant types of interaction between the left and right frontal lobes as indexed by AEC in the δ frequency band, whereas Appendix A shows the dICM between the frontal and temporal lobes bilaterally (indexed by PAC between the θ phase and γ_2_ amplitude). The relative importance of dynamic connectivity measures as (cross-sectional) markers of normal brain development was further assessed through machine-learning techniques (Section 3.2).

### 3.2. Identifying Age-Related Neuromagnetic Features 

Support vector regressors (SVR) were employed in order to estimate the degree of predominance of each dICM (indexed by corresponding MSS and FO values) as a correlate of participant age. Results were very similar across cross-validation schemes: As shown in Appendix A, the linear combination of the 22 dICM features accounted for a substantial portion of age variance in the dataset (*R*^2^ = 0.893, *p* < 2 × 10^−9^; see Appendix A for weights of individual metrics). 

In a second set of analyses, multi-class support vector machines were applied to the same dataset (MSS and FO values) to classify participants into one of six age groups (8–12, 13–17, 18–27, 28–37, 38–50, 51–60 years). The best classification performance was achieved by the 5-fold method, averaging 89.12 ± 5.45% (see Appendix A). Table 1 ranks the dICM features used to classify participants to the correct chronological age group, according to MSS and FO values.

In comparison, the combination of features derived from supplementary metrics (Relative Power [RP], Imaginary Coherence [ImCOH], and Multiscale Entropy [MSE]) was associated with considerably lower classification accuracy (69.05 ± 7.15%) and percentage of age variance accounted for (*R*^2^ = 0.812, *p* < 1.8 × 10^−7^; Appendix A). Among the latter, four features of lobe-averaged MSE displayed the best age prediction results (*R*^2^ = 0.714, *p* < 2.9 × 10^−6^, see Appendix A). These features reflected entropy in signals recorded over the right temporal and parietal areas in the theta and gamma bands, respectively. Static measures of brain activity were clearly inferior to both dICM metrics and MSE in predicting participant age (RP: *R*^2^ = 0.430, *p* < 3.1 × 10^−4^, Appendix A; imCOH: *R*^2^ = 0.525, *p* < 1.9 × 10^−4^, Appendix A). 

### 3.3. Maturation Patterns

Global flexibility index (FI) values were computed for each participant quantifying the frequency of transitions to a different dICM between consecutive time windows of neuromagnetic data. Model comparison using the Akaike information criterion suggested that the FI = a × *AGE^2^* + b × *AGE* + c equation ensured the best fit of individual FI values to participant age (*r*^2^ = 0.88, permutation test *p* < 0.001). Figure 3A demonstrates the rapid growth of FI values through approximately 40 years followed by a gradual decline thereafter. 

### 3.4. Reliability and Clinical Validity of the Flexibility Index

The equivalence of derived dICM values, as correlates of chronological age, across the two MEG systems was supported by two lines of evidence. First, the maturation patterns of subnetwork topography were very similar across systems as illustrated in Appendix A. Second, the model-fit results of the flexibility index values over chronological age were also very similar across MEG systems (see Appendix A). Furthermore, flexibility indices were very stable over time as indicated by the one-week test–retest data from a small dataset of 10 healthy young adults (see Figure 3B and Appendix A). 

Importantly, FI values were found to be consistently lower in school-age children with reading disability (*N* = 25) and adults tested in the acute phase following mild traumatic brain injury (*N* = 30; see Figure 3B) than the age-matched typical participants (Wilcoxon Rank Sum Tests: *p* = 1.63 × 10^−6^ and *p* = 2.91 × 10^−6^, respectively).

## 4. Discussion

Our analyses permitted identification of the spatial pattern and types of the most prominent interactions between underlying neuronal populations that displayed significant developmental trends. Table 1, Figure 2 and Appendix A reveal widespread inter- and intra-hemispheric interactions (e.g., bilateral frontal, parieto-occipital, occipital, temporal, and frontotemporal) involving amplitude envelope correlation, phase synchronization, and phase-amplitude modulation. Cross-sectional developmental curves featured notable peaks in late adolescence and early adulthood, a finding consistent with the notion of protracted maturation of fine-grained cortical synchrony (e.g., [54]). To our knowledge, this is the first demonstration of such developmental trends in resting-state neuromagnetic recordings, which are uniquely suited to model distinct types of population-level interactions involving neuronal signalling in real time. Neural oscillations depend on anatomical and physiological parameters that undergo significant changes during development including changes in GABAergic interneuron activity [55]. These interneurons play a pivotal role in establishing neural synchrony in local circuits, as indicated by the fact that a single GABAergic neuron may be sufficient to synchronize the firing of a large population of pyramidal neurons [56] and that the duration of the inhibitory post-synaptic potential (IPSP) can determine the dominant frequency of oscillations within a network [57].

Awaiting data directly linking dICM measures with individual age-adjusted cognitive abilities, the current demonstration of increasing strength of interactions between fronto-parietal neuronal populations during late childhood is consistent with previous resting-state fMRI reports of increasing functional connectivity between frontal and parietal brain areas through early adulthood [14]. Such developmental trends may be related to the reported strengthening of top-down frontal cognitive control networks [58]. In a similar vein, the rising significance of distinct fronto-temporal dICMs in the repertoire of cortico-cortical interaction modes during the same age range may parallel the continuing functional specialization of fronto-temporal networks supporting memory and executive functions [59]. 

The present findings bear particular relevance to two topics that have attracted growing attention in neurophysiology in recent years, namely the importance of cross-frequency, phase-to-amplitude (CFC^PAC^) interactions between neuronal populations, and the relevance of temporal variability in neuronal coupling modes at rest for brain function. 

Phase-to-amplitude coupling appears to serve a crucial role in the coordination of processes that take place in remote neuronal populations, each operating at different characteristic frequencies [60]. In the present data, CFC^PAC^ was the predominant mode of fronto-temporal neuronal interactions undergoing significant development throughout late childhood and into early adulthood, consistent with their proposed crucial role for information encoding, inhibition, and hierarchical organization of cortical systems [15]. This finding represents a significant advance in the study of cortical synchronization and communication by stressing the developmental significance of phase-amplitude interaction modes that have only recently been introduced to supplement amplitude envelope correlation and phase coupling (e.g., [61]). 

The dynamic connectivity approach adopted in the present study permits quantification of the degree of short-term consistency of various intrinsic coupling modes in time. Results showed that, in addition to systematic developmental changes in the repertoire of dominant interactions that characterize cortico-cortical communication on a global scale, the degree of temporal fluctuation of each type of interaction during the recording session was a robust correlate of age. Moreover, developmental reading disability associated with aberrant brain organization ([28,31] as well as acute brain insult without visible structural damage (i.e., mild traumatic brain injury [19,62,63,64,65]), were characterized by lower temporal variability in dynamic functional connectivity than expected based on the affected persons’ age. According to an emerging view, functional network flexibility may reflect the adaptive capacity of the brain both in short-term situations (i.e., during acquisition of a new skill), and in the course of development [52]. It should be noted, however, that aberrant dynamic features of cortical interactions were common to the two clinical groups and, therefore may not be linked to disease-specific pathophysiological processes. 

## 5. Conclusions

The current results highlight the (i) presence of systematic profiles of functional connectivity between remote cortical areas at rest, and (ii) the potential significance of dynamic flexibility in coupling models for brain maturation. Future studies are needed in order to quantify the range of temporal variability in dynamic functional connectivity that may be optimal for cognitive development and expand the age range covered by the present report to older participants. In view of the considerable, demonstrated sensitivity of neuromagnetic resting-state data to individual participant characteristics such as age, future studies are forthcoming to explore aberrant synchronization patterns of neural oscillations at the cortical source level that may be causally linked to developmental disorders (such as dyslexia, e.g., [28,31]), traumatic brain injury ([19], and disorders that display a notable age-related onset peak (such as schizophrenia [66]). 

## Figures and Tables

**Figure 1 brainsci-09-00380-f001:**
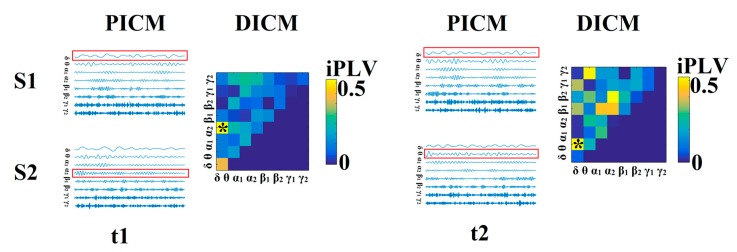
Determining dominant intrinsic coupling modes (dICMs) between two sensors (S1, S2) for two consecutive 2-s sliding time windows (t_1_, t_2_) during the resting-state MEG recording. In this example, the functional interdependence between band-passed signals from the two sensors was indexed by imaginary phase locking (iPLV). In this manner, iPLV was computed between the two sensors either for same-frequency oscillations (e.g., δ to δ) or between different frequencies (e.g., δ to θ). Statistical filtering, using surrogate data for reference, was employed to assess whether each iPLV value was significantly different than chance. During t_1_, the dICM reflected significant phase locking between δ and α_2_ oscillations (indicated by red rectangles) whereas during t_2_, the dominant interaction was found between δ and θ oscillations. Significant values were subsequently integrated over groups of sensors roughly corresponding to underlying lobar anatomy to obtain indices of the dominant type of interaction between hemispheres for a given lobe or between lobes for a given hemisphere. Finally, from the set of potential intrinsic coupling modes (PICM), we derived the dICM for each pair of sensors across all temporal segments. For further details see [29].

**Figure 2 brainsci-09-00380-f002:**
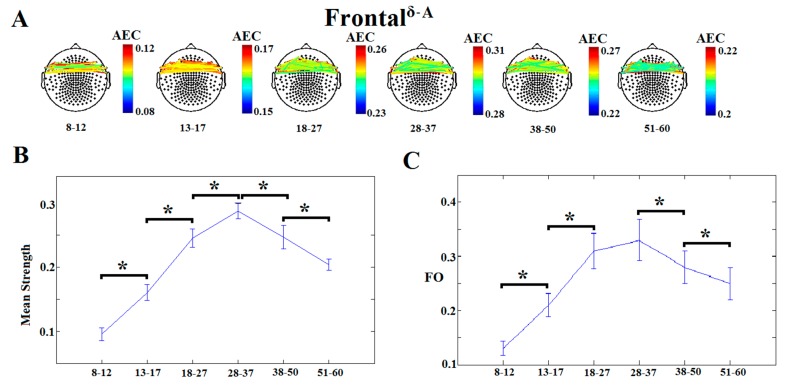
Dominant inter-hemispheric frontal coupling indexed by δ-band amplitude envelope correlation (AEC). (**A**) Topographical layout of statistically significant sensor pairs for the six age groups. (**B**) Mean subgraph strength (MSS) and (**C**) fractional occupancy (FO) derived from envelop correlation across the six age groups. Significant differences between successive age groups are marked by brackets (*p* < 0.0001).

**Figure 3 brainsci-09-00380-f003:**
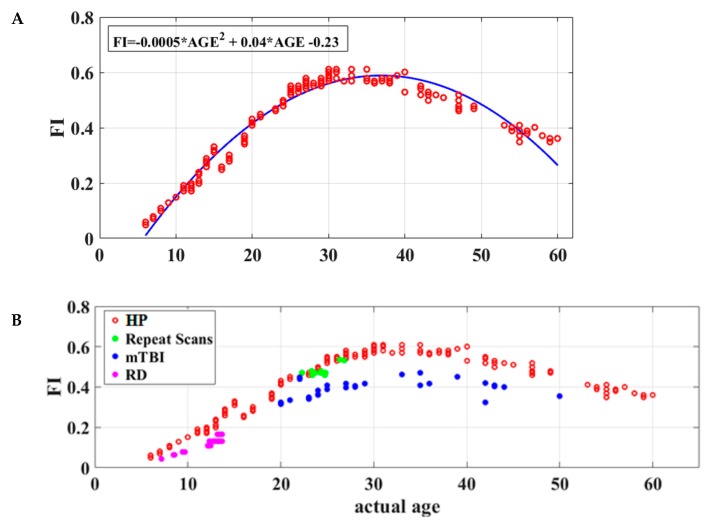
Functional brain maturation curves. (**A**) Individual flexibility index values (FI) for 178 healthy volunteers without history of learning disability or brain injury (aged 6 to 60 years) reflecting the degree of short-term stability of dominant functional connections during the 3 min resting-state recording. Chronological age is shown on the *x* axis. The best-fitting curve for FI as a function of age is indicated by the blue line. (**B**) Flexibility index as a function of age for healthy participants (*n* = 178; HP: red circles), school-aged children displaying severe reading difficulties (*n* = 25; RD: purple circles), adults who had recently suffered mild traumatic brain injury (*n* = 30; mTBI: blue circles), and healthy adults who were retested over a week-long period (*n* = 10; repeat scans: green circles).

**Table 1 brainsci-09-00380-t001:** Ranking of dICM features used to classify participants to the correct chronological age group according to mean subgraph strength (MSS) and fractional occupancy (FO).

Topography of Regional Interaction		Frequency Band	Connectivity Metric	MSS	FO
Frontal	Cross-hemispheric	δ Amplitude	Envelop Correlation	4	20
Frontal–Temporal	Within and cross-hemispheric	θ Phase → γ_2_ Amplitude	Phase-Amplitude Coupling	3	6
Frontal–Parietal	Within and cross-hemispheric	Θ → α_2_ Amplitude	delay Symbolic Transfer Entropy	1	2
Parieto-Occipital	Cross-hemispheric	α_1_ Phase	imaginary Phase Locking	15	14
Frontal	Within hemispheres	θ Phase	imaginary Phase Locking	13	16
L Temporal–Frontal	Cross-hemispheric	δ Phase → β Amplitude	Phase-Amplitude Coupling	5	18
R Temporal–Frontal	Within and cross-hemispheric	δ Phase → γ_2_ Amplitude	Phase-Amplitude Coupling	7	-
L Parietal–Parieto-Occipital	Within and cross-hemispheric	A_1_ Phase	imaginary Phase Locking	12	17
Parieto-Occipital	Cross-hemispheric	β Amplitude	Envelope Correlation	8	21
R Temporal-Parieto-Occipital	Within and cross-hemispheric	γ_1_ Phase	imaginary Phase Locking	10	11
Temporal	Cross-hemispheric	β Amplitude	Envelope Correlation	9	22
Occipital	Cross-hemispheric	α_2_ Phase → γ_1_ Amplitude	Phase-Amplitude Coupling	19	-

L: left, R: right. Unless otherwise specified, indices were integrated over hemispheres.

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
