# Peer review of "Typical and Aberrant Functional Brain Flexibility: Lifespan Development and Aberrant Organization in Traumatic Brain Injury and Dyslexia"

_brainsci, 2019, doi:10.3390/brainsci9120380_

Round 1

Reviewer 1 Report

SUMMARY

This paper deals with developmental changes and differences between typical and atypical functioning in patterns of brain activity at rest. Brain activity was characterized by couplings between patterns of rhythmic fluctuations, essentially the “Dominant Intrinsic Coupling Modes” (dICMs) consisting in amplitude envelope and phase synchronization. The main objectives of the study were (1) to identify characteristic dICMs that demonstrate systematic developmental trends and (2) to develop a measure of temporal variability in dICMs which could serve as a reliable indicator of participant age. Among secondary goals, the most salient one to assess the sensitivity of dICMs in differentiating between typical and atypical brain function. The main group of participants included 178 healthy adults (in 6 age groups between 8 and 60 yrs.). Other groups included 25 children with reading disability (12 yrs. mean age) and 30 adults who had suffered mild traumatic injure (29 yrs. mean age). MEG signals were collected and underwent successive processing stages in order to extract 22 different dICMs indexes. Results comply with the expectations.

MAIN COMMENTS

This paper gives interesting results about the effects of age and neural disorders on couplings between brain networks. However, there is an imbalance between the fairly sophisticated indexes that are used to characterize the couplings and the elementary factors that are used as predictors (age, presence/absence of a disorder whatever its nature). Most importantly, the description of the neuronal indexes is difficult to follow and to relate to the presentation of the results (see further comments below).

Objectives:

The idea to use dICMs as indicator of participant age is a bit awkward. Using age as an indicator of dICMs would make more sense.

Data Processing:

The mathematical processing of the MEG signals is highly complex and presented in a lengthy description that takes about 10 pages of the paper (on a total of 24 including the references. This section should be considerably reduced – perhaps sending parts to the supplemental paper- and clarified. For instance, it is not clear how12 dICMs are represented by 12 MSS and 10 FO values (line 534), how the latter are related to 11 MSS and 11 FO in Figs. S15 and S18, etc.

Results:

Comparing the two following statements, one might wonder how the same set of variables reach about the same performance to classify participants into age groups and to predict individual ages. Explained individual variance should be lesser than correct classification into groups. Why is this not the case here?

Line 569: “ …the linear combination of the 22 dICM features accounted for a substantial portion of age variance in the data set (R2=0.893 …”

Line 571: “In a second set of analyses, Multi-class Support Vector Machines were applied to the same data set (MSS and FO values) to classify participants into one of six age groups (8-12, 13-17, 18-27, 28-37, 38-50, 51-60 years). The best classification performance was achieved by the 5-fold method, averaging 89.12 ± 5.45% …”

The difference between groups should be tested appropriately (add P value).

Line 600:” Importantly, FI values were found to be consistently lower in school-age children with reading disability (N=25) and adults tested in the acute phase following mild traumatic brain injury (N=30; see Fig. 4 lower panel) than age-matched typical participants.”

DISCUSSION

The fact that both children with dyslexia and adults who had suffered mild traumatic brain injury exhibit lower temporal variability in dynamic functional connectivity suggest that the latter is a general index of atypical brain functioning, that is not specific to some disorder. This point should be commented in the discussion.

Author Response

Reviewer 1:

This paper gives interesting results about the effects of age and neural disorders on couplings between brain networks. However, there is an imbalance between the fairly sophisticated indexes that are used to characterize the couplings and the elementary factors that are used as predictors (age, presence/absence of a disorder whatever its nature). Most importantly, the description of the neuronal indexes is difficult to follow and to relate to the presentation of the results (see further comments below).

Objectives:

The idea to use dICMs as indicator of participant age is a bit awkward. Using age as an indicator of dICMs would make more sense.

Answer: While, conceptually, predicting a demographic variable such as age may appear unorthodox, in our analyses this convention is used merely for statistical/computational purposes following a similar trend in the related literature (e.g., Dosenbach et al., Science, 2010; Qin et al., Front. Hum. Neurosci, 2015). The ultimate goal of our analyses as stated in the last paragraph of the Introduction was to “..model age-related individual differences….(1) identify characteristic dICMs within and between lobar regions that demonstrate systematic developmental trends, (2) develop a measure of temporal variability in dICMs, integrated across the entire network of MEG sensors, which could serve as a reliable indicator of participant age”.

Data Processing:

The mathematical processing of the MEG signals is highly complex and presented in a lengthy description that takes about 10 pages of the paper (on a total of 24 including the references. This section should be considerably reduced – perhaps sending parts to the supplemental paper- and clarified. For instance, it is not clear how12 dICMs are represented by 12 MSS and 10 FO values (line 534), how the latter are related to 11 MSS and 11 FO in Figs. S15 and S18, etc.

 Answer: Several sections of text containing technical computational details have been moved to the Supplementary Materials file.

We detected 12 dICMs in total, which in all further analyses were represented by 22 features (12 MSS indices computed for within- and between-sensor associations, and 10 FO indices which, by definition, were computed only for between-sensor associations). This is clarified in p. 11 of the revised manuscript. Color-coding in Fig S15 has been corrected to accurately reflect MSS and FO features.

Figure S18 illustrates the best 12 (independently derived) relative power features derived from Frontal (F) and Parieto-Occipital sensors (PO).

Results:

Comparing the two following statements, one might wonder how the same set of variables reach about the same performance to classify participants into age groups and to predict individual ages. Explained individual variance should be lesser than correct classification into groups. Why is this not the case here? Line 569: “ …the linear combination of the 22 dICM features accounted for a substantial portion of age variance in the data set (R2=0.893 …” Line 571: “In a second set of analyses, Multi-class Support Vector Machines were applied to the same data set (MSS and FO values) to classify participants into one of six age groups (8-12, 13-17, 18-27, 28-37, 38-50, 51-60 years). The best classification performance was achieved by the 5-fold method, averaging 89.12 ± 5.45% …”

Answer: Although variance accounted for by a linear combination of variables and classification performance are derived independently and though different computational procedures, they are typically related strongly to each other. In the present case, it so happened that the respective estimates were also numerically similar.

The difference between groups should be tested appropriately (add P value).  Line 600:” Importantly, FI values were found to be consistently lower in school-age children with reading disability (N=25) and adults tested in the acute phase following mild traumatic brain injury (N=30; see Fig. 4 lower panel) than age-matched typical participants.”

Answer: p values associated with pairwise Wilcoxon Rank Sum Tests have been added in the text as suggested.

DISCUSSION

The fact that both children with dyslexia and adults who had suffered mild traumatic brain injury exhibit lower temporal variability in dynamic functional connectivity suggest that the latter is a general index of atypical brain functioning, that is not specific to some disorder. This point should be commented in the discussion.

Answer: This is an important point that is stressed in the revised Discussion as follows: “It should be noted, however, that aberrant dynamic features of cortical interactions were common to the two clinical groups and, therefore may not be linked to disease-specific pathophysiological processes.”

Reviewer 2 Report

In the present paper it has been reported results from a dynamic functional connectivity approach to determine dominant intrinsic coupling modes in resting-state from MEG. 178 healthy participants aged 8-60 years were tested. Results revealed significant developmental trends in three types of dominant intra- and inter-hemispheric neuronal population interactions involving frontal, temporal, and parieto-occipital brain areas. Moreover, the results showed a systematic temporal variability in functional connectivity profiles displaying an inverse U-shaped curve among healthy participants. Lower flexibility values were found among age-matched children with reading disability and adults who had suffered mild traumatic brain injury. The authors concluded about the importance of these results for normal and abnormal brain development in the light of the recently proposed role of cross-frequency interactions in the fine-grained coordination of neuronal population activity that can be of paramount importance in clinical practice.

I was delighted in reading the present study: the results are intriguing, and the methodology used was strong. My only major concern is about the presentation of the results: in my opinion, the paper is difficult to read because of several technical specifications, and the results the authors reported are really a lot (too much?). It would be more appropriate to concentrate on one, at least two, specific aims in order to better report these interesting results, in order to permit to the reader a more fluent reading and understanding of this huge work.

Author Response

Reviewer 2 

My only major concern is about the presentation of the results: in my opinion, the paper is difficult to read because of several technical specifications, and the results the authors reported are really a lot (too much?). It would be more appropriate to concentrate on one, at least two, specific aims in order to better report these interesting results, in order to permit to the reader a more fluent reading and understanding of this huge work.

Answer: We thank you for your positive feedback. To streamline presentation of the rather complex analysis pipeline, we have moved a considerable portion of the computational details to the Supplementary Materials file. Although, group comparisons and assessment of test-retest reliability are not directly linked to the main research goal (identifying dynamic connectivity metrics that vary with participant age), these analyses are important in order to support the validity of the proposed analysis pipeline. This is more of an issue given that the pipeline as a whole has not been described (and applied to real data) elsewhere.

Reviewer 3 Report

This was a fascinating and thorough study of different network metrics and their integration to explore near-instantaneous fluctuations of brain connectivity and to fingerprint those changes as a function of normal aging as well as in impaired states such as reading difficulties in children and patients with acute mTBI. I expect that the combination of these methods will prove to be fruitful in the exploration of brain function in a variety of diseases in the future. 

I only have minor comments about the manuscript as it stands.

First, is handedness an issue in relation to your network metrics? The connectivity might change a bit in the case of left-handed participants, especially since language dominance can be right, bilateral, or even left sided, and this might alter the profiles of temporal or parietal lobe connectivity. Similarly, were the participants with reading difficulties all right-handed?

What software was used for your analysis? The supplement indicates the use of MatLab and some other programs, and this might be useful to provide in the main document.

dICM should be defined in line 83

There are some minor typos in the body of the document as well as in some of the figures, and some of the abbreviations. Also, the font sizes are inconsistent.

Author Response

Reviewer 3

I only have minor comments about the manuscript as it stands. First, is handedness an issue in relation to your network metrics? The connectivity might change a bit in the case of left-handed participants, especially since language dominance can be right, bilateral, or even left sided, and this might alter the profiles of temporal or parietal lobe connectivity. Similarly, were the participants with reading difficulties all right-handed?

Reply: All participants with dyslexia and mTBI were selected for this analysis to be right-handed. There were 5 left-handed participants reported in the open data bases of MEG data who were not included in the analyses.

What software was used for your analysis? The supplement indicates the use of MatLab and some other programs, and this might be useful to provide in the main document.

Reply: As noted in p. 14 of the revised MS: “All analyses were conducted using custom in-house software in MATLAB 2018b and Fieldtrip basic routines for reading MEG files.”

dICM should be defined in line 83

Reply: dICM is defined when it first appears in the text on p. 2

There are some minor typos in the body of the document as well as in some of the figures, and some of the abbreviations. Also, the font sizes are inconsistent.

Reply: We read carefully through text to correct typos and inconsistencies.

Round 2

Reviewer 1 Report

I saw that the authors complied with most of my comments. The paper is now much clearer and accessible for readers interested in mathematical developments in neurosciences